# SELFCHECK: USING LLMS TO ZERO-SHOT CHECK THEIR OWN STEP-BY-STEP REASONING

**Ning Miao**[1*]    **Yee Whye Teh**[1]    **Tom Rainforth**[1]

## ABSTRACT

The recent progress in large language models (LLMs), especially the invention of chain-of-thought prompting, has made it possible to automatically answer questions by stepwise reasoning. However, when faced with more complicated problems that require non-linear thinking, even the strongest LLMs make mistakes. To address this, we explore whether LLMs are able to recognize errors in their own step-by-step reasoning, without resorting to external resources. To this end, we propose SelfCheck, a general-purpose zero-shot verification schema for recognizing such errors. We then use the results of these checks to improve question-answering performance by conducting weighted voting on multiple solutions to the question. We test SelfCheck on math- and logic-based datasets and find that it successfully recognizes errors and, in turn, increases final answer accuracies.

## 1 INTRODUCTION

Recent years have witnessed dramatic changes in the areas of NLP and AI brought on by significant advances in LLMs. From GPT-3 (Brown et al., 2020), PaLM (Chowdhery et al., 2022), Llama (Touvron et al., 2023) and Falcon (Almazrouei et al., 2023) to GPT-4 (OpenAI, 2023) and PaLM-2 (Google, 2023), the increasing model sizes and exploding amount of training data have empowered LLMs to achieve human-level performance on a large range of tasks, including summarization, translation, and question answering. The invention of Chain-of-Thought prompting (CoT, Wei et al. (2022)) has further enhanced LLMs' ability to solve complex problems by generating step-by-step solutions.

However, the performance of even the largest LLMs is still unsatisfactory on more difficult reasoning problems. For example, GPT-4 with CoT prompting only correctly answers 42.5% of problems in the MATH dataset (Bubeck et al., 2023; Hendrycks et al., 2021), which is far below the human level. Such problems require careful multi-step reasoning to solve, and LLMs are consequently prone to make mistakes: even though their error rate on individual steps may be low, the probability of generating at least one erroneous step can still be quite high, undermining the final answer.

Recent works have tried to overcome this limitation by checking for errors in these step-by-step solutions (Cobbe et al., 2021; Li et al., 2022; Ling et al., 2023; Jiang et al., 2023). Such checks can then be used to provide confidence scores in answers and select between different possible alternatives. This checking has typically been performed either by using an external verification model (Cobbe et al., 2021; Lyu et al., 2023; Peng et al., 2023), or through few-shot in-context learning (Brown et al., 2020) of an LLM (Weng et al., 2022; Ling et al., 2023). Unfortunately, existing methods generally require extra training data and/or domain-specific exemplars, which often makes them inconvenient to use in practice and restricts them to specific domains or data formats. The aim of our work is thus to instead provide a general-purpose, zero-shot, approach to checking that relies only on the original LLM, without the need for additional external resources.

To this end, we introduce SelfCheck, a zero-shot step-by-step checker for self-identifying errors in LLM reasoning chains. SelfCheck uses the LLM to individually check the conditional correctness of each step based on directly related information from the question and the preceding steps, in a manner similar to a human going back to check their work. The results of these individual checks are then integrated to form an overall correctness estimation for the whole reasoning chain.

Key to SelfCheck's success is a novel mechanism for performing the checking of individual steps. As we will show, the naive approach of directly asking the LLM to check a step is typically ineffective.

---

[1]Department of Statistics, University of Oxford. [*]Email: <ning.miao@stats.ox.ac.uk>.

Instead, we introduce a multi-stage approach that breaks the problem down into a series of simpler tasks, leverages the generative strengths of the LLM, and decorrelates errors between the original generation and checking. Specifically, using separate calls to the LLM we first extract the target and relevant context for the step, then regenerate an independent alternative step from these, and finally compare the two. The original step is then deemed to pass the check if it matches the regeneration.

Besides providing an estimation of correctness for each solution, SelfCheck can also boost final answer accuracies for the original questions by weighted voting. Namely, given multiple solutions to a question, it uses confidence scores as weights to vote among the answers, which provides a soft way to focus on more accurate solutions.

We evaluate SelfCheck on three math tasks and one logical reasoning task. For all datasets, we find that using SelfCheck achieves a significant increase in final answer accuracies compared with simple majority voting and other baselines. We also see that SelfCheck provides an accurate confidence estimation for LLM's solutions, which decreases the proportion of incorrect solutions by between 9% and 23% (depending on dataset) when filtering out solutions with low confidence scores. We further perform a number of ablations to justify some of our key design choices in the SelfCheck approach.

To summarize, we introduce SelfCheck as a novel and effective zero-shot schema for self-checking step-by-step reasoning in LLMs. Unlike previous methods, SelfCheck does not need any finetuning or example crafting, so can be directly applied to reasoning tasks in different domains. Our experiments confirm that it can, in turn, be used to improve final predictive performance of LLMs.

## 2 RELATED WORK

How to automatically check the correctness of a sequence of reasoning steps is a long-standing question. We now discuss how previous methods have tried to tackle this in an LLM context. We note that none of these works are able to work in the zero-shot setting covered by SelfCheck, requiring either problem-specific examples, an external model, and/or finetuning.

**Few-shot verification** Though our focus will be on zero-shot checking, for some problems one may have hand-crafted exemplars available that are specifically designed for that particular question-answering task. Previous methods have been designed to perform checking of LLMs' generated solutions in this few-shot checking scenario.

For example, the Self-Verification (SV) approach of Weng et al. (2022) verifies the whole solution by backward prediction. That is, it uses the conclusion from CoT reasoning to predict a masked condition in the question. However, it only supports single-step checking and is based on the assumption that every piece of information in the question can be recovered using a correct solution of it, which is often not the case. Consequently, it is only applicable to simpler tasks, such as GSM8K.

The Deductive Verification (DV) approach of Ling et al. (2023) instead looks to verify independent sub-tasks, as per SelfCheck. However, its verifier only supports checking reasoning chains in a special format called Natural Programs. As a result, it can only work with a specific specialised generator, without serving as a general verifier for multi-step reasoning.

**Verification with external resources** In some cases, there might be external resources available to verify the logical correctness or faithfulness of LLM outputs. Lyu et al. (2023) translate a question into a symbolic reasoning chain using an LLM and solve the problem by a symbolic logic solver. Peng et al. (2023) introduced an external database to check for incorrect knowledge in LLM outputs. These methods are limited by the availability of external resources and are typically restricted to checking for certain types of errors.

**Training/finetuning a verifier** A few other methods train or finetune a separate verifier model to check reasoning chains. Cobbe et al. (2021) finetuned a GPT-3 model on GSM8K to predict the correctness of a solution as a whole. Li et al. (2022) trained a binary *deberta-v3-large* (He et al., 2020) classifier on each domain to predict step correctness. More recently, Lightman et al. (2023) built a large dataset, which contains step-wise correctness labels from human labelers, and finetuned a GPT-4 model on it. Unlike SelfCheck, all of these methods require extra data and external computational resources, restricting their applicability and ease of use.

# 3   SELFCHECK: USING LLMs TO CHECK THEIR OWN REASONING

Rather than relying on external resources or problem-specific data like the aforementioned approaches, it would be highly beneficial if we could develop self-contained checking schemes that require only the original LLM itself. In other words, we would like to use the LLM to identify errors in its own step-by-step reasoning, analogously to how a human might go back to check their working.

Unfortunately, directly asking the LLM to check its own reasoning is largely ineffective: it almost invariably declares that the original answer is correct, with Ling et al. (2023) finding answers checked in this way are deemed correct more than 90% of the time regardless of whether they actually are. As we will show in Section 5, individually prompting the LLM to check each step in the CoT reasoning fares slightly better, but is still only able to offer marginal gains compared to not checking at all.

A more nuanced method to perform this checking is thus required. To this end, we introduce **SelfCheck**, a general-purpose, zero-shot, checking schema for self-identifying errors in LLM CoT reasoning. Given a question, $q$, and its step-by-step solution, $s$, produced by some **generator** (which will generally be an LLM with appropriate CoT prompting), SelfCheck considers each step of $s$ in turn and tries to establish its individual correctness based on the preceding steps. This checking is done by leveraging an LLM (which can either be the same LLM used to generate $s$ or a separate one), but rather than directly asking the LLM to perform the check, we instead introduce a novel step checking method (see Section 3.1) that exploits their generative modeling strengths. The results of the checks on individual steps are then combined into a single confidence score, $w \in [0, 1]$, for the whole solution. These confidence scores, in turn, allow us to improve predictive performance, by using them to perform weighted voting on multiple solutions to the same question.

## 3.1   STEP CHECKING

To check individual steps of reasoning process, the first thing we should note is that the correctness of each step is highly dependent on its context, namely the question and previous steps in the solution. For example, we usually need to refer to previous steps for the definition of variables and the meaning of specific numbers. If each step is conditionally correct based on the provided context and the last step provides an answer in the required format, then the overall reasoning will itself be correct. The target of the step checking is thus simply to check the conditional correctness of each step based on the context provided by previous steps. That is, we only care about catching errors at the current step, and can assume all information from its context to be correct.

A simple idea to achieve this would be to feed the current step as well as all its context to an LLM and directly ask it to 'check the correctness of the step'. However, in practice, we find that this task is too difficult for current LLMs to do effectively, even with careful prompting that exemplifies how to do the checking in detail (see Section 5). This difficulty comes first from the fact that there are multiple aspects to the checking problem that the checker must deal with simultaneously: it needs to understand the key content in the step and then collect all related information from the context, before actually checking for its correctness. Second, LLMs are trained as generative models, rather than directly as supervised approaches for checking, such that it is a problem that does not necessarily play to their strengths. Finally, there are likely to be strong correlations between the errors such a checker will make with the errors made in the original generation, undermining its usefulness.

To address these difficulties, SelfCheck instead decomposes the checking task for each step into four stages: *target extraction*, *information collection*, *step regeneration*, and *result comparison*. The LLM is used to execute each stage successively, with the outcome of the result comparison providing the correctness prediction.

The idea behind this decomposition is to make the LLM focus on an easier task at each stage and ensure the individual tasks carried out are more closely aligned to the LLM's strengths. Moreover, by focusing on regenerating and then comparing, we hope to reduce the correlations between the errors of the checking and the original generation.

At a high level, the stages work by first prompting the LLM to figure out the target of the current step and what information it uses to achieve the target; we find that the LLM is usually able to perform these tasks extremely accurately. Then we ask the LLM to re-achieve the target using only the collected information, providing an alternative to the original step that maintains the same purpose

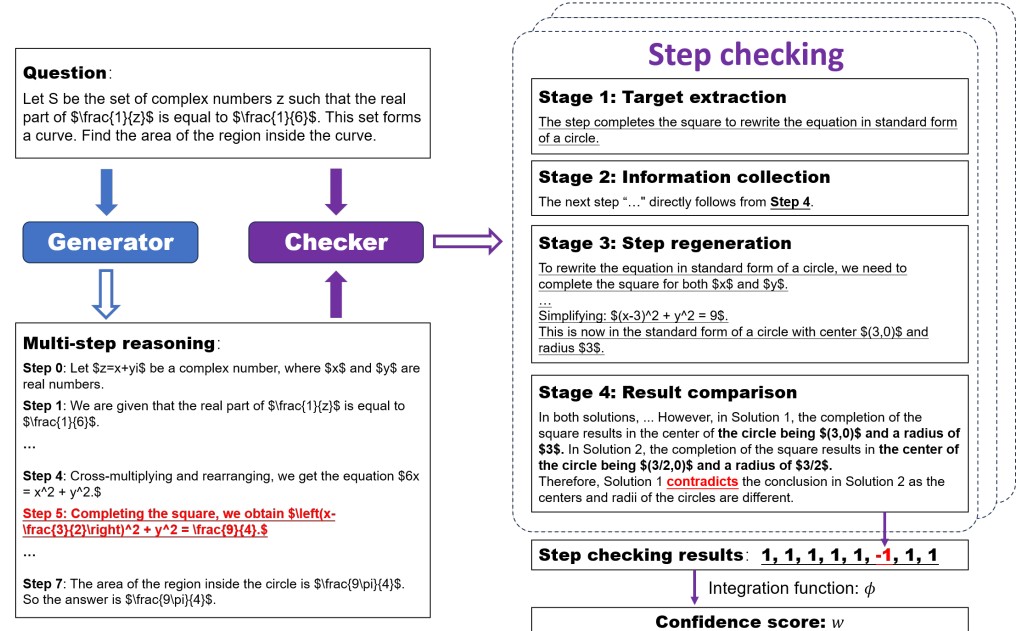

Figure 1: Example of using SelfCheck, focusing on the checking of a particular step (Step 5). To check the correctness of the step, SelfCheck goes through 4 stages. First, in the target extraction stage, it figures out that the main purpose of Step 5 is to complete the square. In the information collection stage, it then establishes that Step 5 only directly relies on Step 4. Next, the step regeneration stage instructs the LLM to complete the square independently, only using Step 4 as context. The regeneration result shows that the center and radius of the circle are $(3, 0)$ and 3, which is different from what is implied by the original Step 5. Consequently, the result comparison stage concludes that Step 5 is likely to be wrong. After checking all the steps, SelfCheck integrates the results to form an overall confidence score, $w$. See Appendix A for a complete version of the example.

in the overall reasoning process. Here the clear description of the target and the simplified context we provide make the regeneration stage less challenging. As a result, we hope its output will be more reliable and thus serve as a useful reference. Even if this is not the case, it will still hopefully provide a viable alternative, with a distinct generation, that can be used for comparison. The last stage then uses the LLM to compare the original step with the regenerated output. If their main conclusions match/mismatch, this provides evidence that the original step was correct/incorrect.

A worked example of this step-checking process is provided in Figure 1. In the following, we describe each of the subtasks in detail and provide our specific instructions to the LLM. We note here that the different LLM queries are made independently, rather than keeping the queries and answers from previous stages in context. Thus, for example, when the LLM is called to carry out the step regeneration, it does *not* have access to the original generation. The same prompts are used across LLMs and datasets, thereby providing a general-purpose approach.

**Target extraction** To check a step (for example, Step 5 in Figure 1), we first need to figure out what the step is trying to achieve. Without a specific target, the regeneration stage would proceed in a random direction, making it impossible to serve as a reference to the original step. We thus use the LLM itself to extract the target of a step using the question and all previous steps (Steps 0-4 in Figure 1) with the following prompt (we omit some line breaks due to space limitations):

*The following is a part of the solution to the problem [Question]: [Step 0,..., Step i]. What specific action does the step [Step i] take? Please give a brief answer using a single sentence and do not copy the steps.*

During execution, we copy the question and steps into [Question] and [Step 0, ..., Step i] to form the actual input to the LLM. The reason for requesting a brief answer is to try and keep the amount of information retained to the minimum needed, thereby avoiding unnecessary influence on the regeneration and hopefully reducing correlations in errors in turn.

**Information collection** To reduce the difficulty of the regeneration stage and avoid unrelated information from affecting the result, we filter out information that is not directly related to the

current step. Specifically, we ask the LLM to select useful items from the question and all previous items with the following prompt, where [Information j] is simply the j-th sentence in the question:

*This is a math question: [Question]. The following is information extracted from the question:*
*Information 0: [Information 0]   Information 1: [Information 1]   ...*
*The following are the first a few steps in a solution to the problem:*
*Step 0: [Step 0]   Step 1: [Step 1]   ...   Step i-1: [Step i-1]*
*Which previous steps or information does the next step [Step i] directly follow from?*

After retrieving the free-text response from the LLM, we extract step or information ids by regular expression. For example in Figure 1, the current step requires Step 4 and no information from the question as context. The selected steps and information are then fed into the regeneration stage.

**Step regeneration**   Given the target and necessary information of the step, we can now ask the LLM to achieve the target independently with only the collected information, without seeing the original step. Because the step is usually a small jump from previous conclusions, and the information collection stage has already filtered out irrelevant information, we can usually trust regeneration results. The prompt for this stage is:

*We are in the process of solving a math problem. We have some information from the problem:*
*Information 0: [Information $I_0$]   Information 1: [Information $I_1$]   ...*
*The following are some previous steps:   Step 0: [Step $S_0$]   Step 1: [Step $S_1$]   ...*
*The target for the next step is:   [Target]*
*Please try to achieve the target with the information from the problem or previous steps.*

Here [Target] is the output from the target extraction stage. [Information $I_i$] and [Step $S_i$] correspond to the specific items selected by the information collection stage. In Figure 1, only Step 4 and no information from the question is directly related to the current step, so SelfCheck simply copies the content of Step 4 into [Step $S_0$] and removes the block containing [Information $I_i$].

**Result comparison**   The last step is to compare results from the regeneration stage and the original step with the following prompt:

*The following are 2 solutions to a math problem.   Solution 1: [Regeneration output]   Solution 2: [Step i]*
*Compare the key points from both solutions step by step and then check whether Solution 1 'supports',*
*'contradicts' or 'is not directly related to' the conclusion in Solution 2. Pay special attention to the difference*
*in numbers.*

If the regeneration output 'supports'/'contradicts' the original step, we can conclude that the original step is likely correct/incorrect respectively. Sometimes, the correctness of the original step cannot be directly inferred from the regeneration output. For example, when the target is to simplify an equation, then there may be multiple valid solutions. In such cases, we are not sure about the correctness of the original step, which makes 'is not directly related to' the third possible outcome of the check.

## 3.2   RESULTS INTEGRATION

After running step-checking and getting a checking result for each step, we need an integration function $\phi$ to give a confidence score, $w \in [0, 1]$, for the overall correctness of the solution. The input of $\phi$ should be a vector in the form of $[r_0, r_1, ..., r_n]$, where each item $r_i$ represents the step checking result for Step $i$. We will use $r_i = -1, 0$, and $1$ to represent the step-checking results 'contradict', 'is not directly related to' and 'support' respectively. We find that the following simple integration function works well in practice

$$w = \phi([r_0, r_1, ..., r_n]) = 2 * \text{Sigmoid}\left(-\lambda_{-1} \sum_{i=0}^{n} \mathbb{1}_{r_i=-1} - \lambda_0 \sum_{i=0}^{n} \mathbb{1}_{r_i=0}\right), \tag{1}$$

where $\lambda_{-1}$ and $\lambda_0$ are two non-negative hyperparameters with $\lambda_{-1} > \lambda_0$; we found nearly any sensible pairs of $\lambda_{-1}$ and $\lambda_0$ worked similarly well in preliminary tests, so we simply fix $\lambda_{-1} = 1$ and $\lambda_0 = 0.3$ throughout our experiments. The rationale of this setup is that the more failed checks we see, the more likely the overall reasoning process, and thus the final solution, are wrong. Note here that, because the checks are themselves imperfect, we do not necessarily want to immediately reject the whole solution from a single step-check failure, especially for $r_i = 0$ cases. This is why we take a 'soft' approach to the verification with a confidence score. The number of successful checks, i.e. $\sum_{i=0}^{n} \mathbb{1}_{r_i=1}$, is deliberately not included in our integration function as an increased number of successful checks does not actually increase our confidence in the overall solution: shorter reasoning chains are generally preferable to longer ones for a given question and LLM.

Once calculated, the resulting confidence score can be directly used as a weight for voting between different possible solutions. We can thus use SelfCheck to increase the accuracy of an LLM's answers by generating multiple possible solutions, calculating confidence scores for each, and then choosing our final answer through weighted voting.

## 4 EXPERIMENTS

We now run experiments on three math-reasoning datasets to evaluate SelfCheck's effectiveness in checking multi-step reasoning and improving final answer accuracies. Note here that our focus on math-reasoning problems is due to ease of performance evaluation and dataset availability; SelfCheck is directly applicable to other question-answering problems with nominal changes to our prompts. An additional experiment based on a logic reasoning task is provided in Appendix D.

**Datasets** GSM8K (Cobbe et al., 2021), MathQA (Amini et al., 2019), and MATH (Hendrycks et al., 2021) consist of math problems on primary school, middle school, and competition levels, containing 1319, 2985, and 5000 test samples, respectively. For GSM8K and MathQA, we evaluate SelfCheck on the whole test sets. Due to limited resources, we use a subset of MATH test set taken from Ling et al. (2023).[1] Besides the levels of difficulty, the three datasets differ from each other in the following aspects. Firstly, MathQA provides 5 options to choose from for each problem, while GSM8K and MATH have no options. Secondly, GSM8K only has arithmetic problems, while MathQA and MATH contain more diverse problems in geometry, physics, probability, and algebra.

**LLMs** We use GPT-3.5 (gpt-3.5-0301) and GPT-4 (gpt-4-0613) as our LLMs, focusing in particular on the former due to budget restrictions. Note that the same prompts are used for all datasets with both LLMs during evaluation; no dataset-specific customization or tuning has been performed. When devising the prompts, a small number of training samples from MathQA dataset were utilized. An additional experiment using Llama2 (70B, 4-bit, Touvron et al. (2023)) is provided in Appendix E.

**Baselines** We use majority voting (also known as Self-Consistency Decoding (Wang et al., 2022) in the context of CoT reasoning) as our main baseline following Ling et al. (2023) and Lightman et al. (2023). Despite its simplicity, this is still quite a strong baseline in the current literature. In particular, most existing few-shot methods report similar results compared with it (Weng et al., 2022; Ling et al., 2023). We also compare with quoted results from Self Verification (SV, Ling et al. (2023)) and Deductive Verification (DV, Weng et al. (2022)) when possible. We note though that these approaches are not directly comparable to SelfCheck in general, as they require additional exemplars which will often not be available in practice. Despite this, we will find that SelfCheck outperforms them when comparisons are possible.

We omit results from Faithful-CoT (Lyu et al., 2023), because it has already been shown to decrease the accuracies on GSM8K and MATH by 11.8% and 4.2%, respectively compared to majority voting (Ling et al., 2023). It is also impossible for us to compare with training/finetuning based methods such as Lightman et al. (2023), because we have neither access to their finetuned models nor computation resources to repeat their training/finetuning. The significant extra data and resources they require also means their contributions are somewhat tangential to SelfCheck regardless.

### 4.1 FINAL ANSWER CORRECTNESS

Figure 2 shows the performance gains using the confidence scores from SelfCheck to do weighted voting compared with baseline methods. The upper plots show that accuracies of both SelfCheck and majority voting have the same increasing tendency as the number of generated solutions per question increases, which is a result of the variance reduction provided by averaging over more solutions. The bottom plots show the difference in accuracy between the two including the standard error in the estimate. We can see that by allocating higher weights to correct solutions, SelfCheck achieves significantly higher accuracies than majority voting for all solution numbers per question. We also find the improvements of SelfCheck (compared with majority voting) to be higher than Deductive Verification and Self-Verification in their reported settings, despite the use of in-context learning from additional examples. We will perform additional experiments on how performance changes when ensembling over a larger number of solutions in Section 5.1.

---

[1] https://github.com/lz1oceani/verify_cot/tree/main/results/chatgpt3.5/natural_program/MATH_np.json

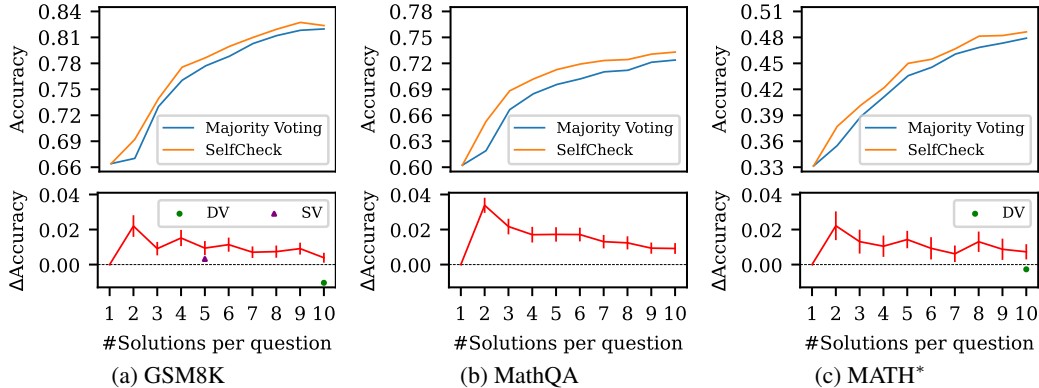

Figure 2: The upper plots show the accuracies of SelfCheck and majority voting for different numbers of generated solutions per question with GPT-3.5. The lower plots show the accuracy gaps between each method and majority voting, where DV and SV stand for Deductive Verification (Weng et al., 2022) and Self-Verification (Ling et al., 2023), respectively. It is difficult to compare with DV and SV with respect to absolute accuracies because they are using different generator models. However, we can see that SelfCheck achieves higher relative performance gains than both in their reported settings.

Table 1: SelfCheck significantly increases final answer accuracies with both GPT-3.5 and GPT-4, even we only have 2 candidate solutions for each question. ΔAcc is the performance gain of SelfCheck compared with majority voting (MV), with the ± indicating the standard error. ✗✗, ✗✓ and ✓✓ represent the proportions of questions with 0, 1 or 2 correct solutions. Unlike in Figure 2, Acc(MV) here is essentially the average accuracy of 2 single generations. We see that the gains from SelfCheck are typically larger in cases where it is common for only one of the solutions to be correct, as these are the cases using weighted voting can influence the final answer.

| Dataset | Generator | Checker | ✗✗ (%) | ✗✓ (%) | ✓✓ (%) | Acc(MV, %) | Acc(SelfCheck, %) | ΔAcc (%) |
|---------|-----------|---------|--------|--------|--------|-----------|-------------------|----------|
| GSM8K | GPT-3.5 | GPT-3.5 | 16.8 | 23.0 | 60.2 | 71.7 | 74.3 | 2.8±0.9 |
| | GPT-4 | GPT-4 | 8.8 | 8.2 | 83.0 | 87.1 | 86.9 | -0.2±0.2 |
| | GPT-4 | GPT-3.5 | 8.8 | 8.2 | 83.0 | 87.1 | **88.1** | 1.0±0.3 |
| MathQA | GPT-3.5 | GPT-3.5 | 27.6 | 26.4 | 46.0 | 59.2 | 64.6 | 5.4±1.1 |
| | GPT-4 | GPT-4 | 16.2 | 11.0 | 72.8 | 78.3 | 80.9 | 2.6±0.4 |
| | GPT-4 | GPT-3.5 | 16.2 | 11.0 | 72.8 | 78.3 | **81.2** | 3.0±0.4 |
| MATH* | GPT-3.5 | GPT-3.5 | 52.6 | 23.2 | 24.2 | 35.8 | 38.0 | 2.2±0.7 |
| | GPT-4 | GPT-4 | 42.0 | 20.2 | 37.8 | 47.9 | **51.3** | 3.4±0.6 |
| | GPT-4 | GPT-3.5 | 42.0 | 20.2 | 37.8 | 47.9 | 48.9 | 1.0±0.8 |

To investigate the effect of using more powerful LLMs, and of using a different LLM for the generation and checking, we further conducted experiments with GPT-4 and a mix of GPT-4 and GPT-3.5. Because of the high cost of calling the GPT-4 API, we randomly sample 500 questions from each dataset to form the test sets and generate 2 (instead of 10) answers to each question. In Table 1, we see that SelfCheck significantly outperforms majority voting with both GPT-3.5 and GPT-4. We also notice that using GPT-3.5 to check GPT-4 generated answers yields surprisingly good results, actually outperforming checking with GPT-4 on the simpler GSM8K and MathQA tasks. This is likely because using different LLMs helps to further decorrelate the errors of the generator and the checker, and shows that using a cheaper LLM can still often be sufficient for the checking. For the more difficult problems in MATH, using GPT-4 as checker always produces better results, but even here the checking from GPT-3.5 is beneficial compared to doing no checking at all.

## 4.2 VERIFICATION PERFORMANCE

Besides serving as a confidence score calculator to improve the performance of voting, SelfCheck can also predict the correctness of a single solution. To do so, we simply set a threshold $t$ to the confidence score, where solutions with confidence scores $w \geq t$ are classified as correct.

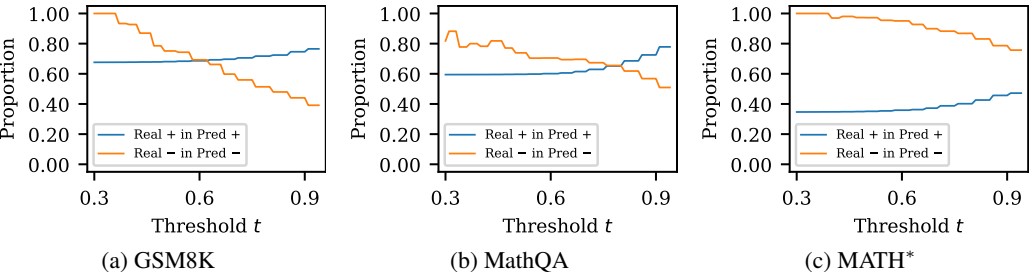

(a) GSM8K      (b) MathQA      (c) MATH$^*$

Figure 3: When raising the classification thresholds $t$, the proportions of real correct solutions in predicted correct solutions (Real + in Pred +) increase for GSM8K (67.5%→76.5%), MathQA (59.4%→82.2%) and MATH (34.6%→50.8%).

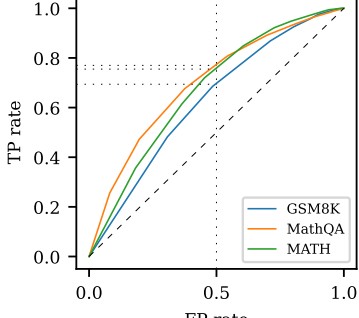

Figure 4 shows the ROC curves for each dataset. As a comparison, directly prompting GPT-3.5 to verify whole reasoning chains leads to no meaningful control on the false and true positive rates (FP and TP): they are always both 100% on MATH and 98% on GSM8K, as observed by Ling et al. (2023). In other words, the checker always predicts the answer as correct, providing no useful information.

As well as verification accuracies, we may also care about the solution quality after filtering out solutions with low confidence scores $w$. Figure 3 shows that by increasing the threshold $t$, SelfCheck can filter out more incorrect solutions, such that a higher proportion of the solutions that pass the check are indeed correct (Real + in Pred +). Though this is at the cost of

Figure 4: True positive rates (TP) vs. false positive rates (FP) as classification threshold, $t$, is varied.

misclassifying more of the real correct solutions as incorrect, this can be a useful feature in cases where the risk of choosing an incorrect solution is higher than rejecting a correct one.

## 5 ANALYSIS

We now perform some ablations to justify some of the key design choices made by SelfCheck and provide insights on its behavior. Limited by budget and time, all experiments in this section are performed on a subset of the MathQA test set with 100 randomly selected questions.

### 5.1 MORE SOLUTIONS PER QUESTION?

Serving as a method to reduce variance, majority voting increased final answer accuracies on different datasets when we increased from 2 to 10 solutions in Figure 2. In cases where we only care about final predictive performance and majority voting can be applied (see Appendix D for an example where it cannot, but SelfCheck can), one might thus question whether it is better to simply use our computational resources to keep increasing the size of this ensemble, rather than relying on checking. Note here that the

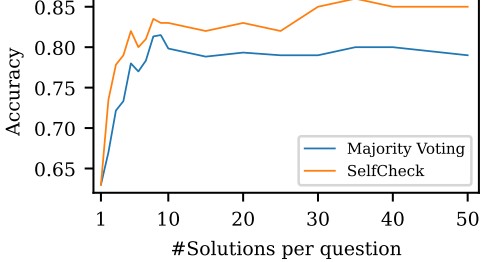

Figure 5: SelfCheck achieves significantly higher final answer accuracies than majority voting for large ensembles of solutions.

cost of running the verifier is around twice that of the original generation for zero-shot generators (it can cost less than the original generation for few-shot generators with longer prompts).

However, as shown in Figure 5, the improving performance from using a larger ensemble saturates for majority voting, with the accuracy never going above that achieved when $n = 9$, thereby never reaching the performance we already achieved by SelfCheck for an ensemble of size $n = 5$. Moreover, the performance of SelfCheck continues to increase as the ensemble grows. By lowering the weights (confidence) of incorrect solutions, SelfCheck increases the chance of selecting the correct answers, even when their generation probabilities in the generator LLM are low. Therefore, with SelfCheck, **LLMs can effectively rectify their own biased beliefs by themselves**.

## 5.2 ABLATION STUDIES

In order to pick apart the effect of several critical design choices for SelfCheck, we compare SelfCheck with some of its variants with respect to final answer and verification accuracies on MathQA.

**Global v.s. step-by-step checking**   The first question is can we simply ask an LLM to check the whole solution without taking steps into consideration. To answer it, we prompt the LLM to perform global checking with the following instruction:

> *The following is a question and a solution to it from a student. Carefully check whether the solution is correct step by step. End your response with your conclusion that starts with "Correct", "Wrong" or "Not Sure".*
> *Question: [Question]    Solution: [Step 0, Step 1,..., Step n]*

Similar to the findings of Ling et al. (2023), we find that the global checker outputs "correct" most of the time and rarely recognizes an error. Consequently, its final answer accuracies are very close to majority voting (in Figure 6) and its verification accuracy (55.0%) is only marginally above random guess (50.0%). This lack of ability to deal with the difficulty of global checking is what makes step checking necessary.

Figure 6: Generation accuracies for variants of SelfCheck on MathQA with GPT-3.5.

**Single-stage v.s. multiple-stage step checking**   Next, we ask whether we really need to decompose the step checking into several stages. To answer this, we design the following prompt to use the LLM directly.

> *The following is a question and the first a few steps in its solution.*
> *Question: [Question]    Solution: [Step 0, Step 1,..., Step i-1]*
> *Check the correctness of the next step: [Step i]*
> *Please consider the information it relies on and check step by step. Please end your response with your conclusion that starts with "Correct", "Wrong" or "Not Sure".*

Figure 6 and Table 2 show that although this is better than global checking, it is still significantly worse than SelfCheck with its multi-stage checking. This indicates that checking a step in a single stage is still too challenging for the LLM, so it is necessary to further decompose step checking into a pipeline of easier sub-tasks.

Table 2: Verification accuracies for variants of SelfCheck on MathQA with GPT-3.5. The reported verification accuracy is the average of true positive and true negative rates.

**Error check v.s. regenerate and compare**   We now justify the choice to perform step regeneration and comparison instead of direct error checking for each step. To do so, we replace our regeneration stage and comparison stage with a single error-checking stage. We first compare with a zero-shot version of the variant with the following prompt:

| Method | Accuracy (%) |
|---|---|
| SelfCheck | **66.7%** |
| Global Check | 55.0% |
| Single stage Check | 57.2% |
| Error Check (0-shot) | 63.1% |
| Error Check (1-shot) | 64.2% |

> *Given the following information:*
> *Information 0: [Information $I_0$]    Information 1: [Information $I_1$]    ...*
> *Step 0: [Step $S_0$]    Step 1: [Step $S_1$]    ...*
> *Check the correctness of the next step [Step i]*
> *Please check for grounding errors, reasoning errors and calculation errors step by step. Please end your response with your conclusion that starts with "Correct", "Wrong" or "Not Sure".*

We then add an examplar from Ling et al. (2023) (see Appendix B) to make a more powerful one-shot error checker. However, results in Figure 6 and Table 2 show that even with a very detailed and instructive example, direct error checking still performs worse than our regenerate and compare approach, which supports our previous argument that LLMs are better at generation than checking.

## 6 CONCLUSIONS

In this paper, we have introduced SelfCheck, a general-purpose, zero-shot, step-by-step checking scheme for LLMs. Unlike previous approaches, SelfCheck does not require any additional data or external resources: it uses the LLM to identify errors in its own reasoning, leveraging a novel regenerate-and-compare approach. By using the results of this checking to perform weighted voting over different solutions, we find that SelfCheck is able to, in turn, increase final predictive accuracy.

ACKNOWLEDGEMENTS

Tom Rainforth is supported by the UK EPSRC grant EP/Y037200/1. Ning Miao and Yee Whye Teh are supported by ELISE European Network of AI Excellence Centres and the European Union's Horizon 2020 research and innovation programme under grant agreement No 951847 as well as by Tencent through the Oxford-Tencent Collaboration on Large Scale Machine Learning. Ning Miao is also supported by China Scholarship Council - University of Oxford Scholarship.

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

## APPENDIX A  A COMPLETE EXAMPLE OF SELFCHECK

To help readers get a better idea of how SelfCheck works, we show a complete version of the example in Figure 1, including the exact inputs and outputs of different stages. Please note that we have compiled all latex codes for readability, though the inputs and outputs to LLMs are in their original form. The question is

*Let $S$ be the set of complex numbers $z$ such that the real part of $\frac{1}{z}$ is equal to $\frac{1}{6}$. This set forms a curve. Find the area of the region inside the curve.*

We feed the following instruction to the LLM that serves as the generator:

*Solve the following problem step by step. Please start each step with "Step :" and split sentences in each step with "\n\n". Please finish you response with "So the answer is ...". Problem: Let $S$ be the set of complex numbers $z$ such that the real part of $\frac{1}{z}$ is equal to $\frac{1}{6}$. This set forms a curve. Find the area of the region inside the curve.*

After getting the following solution, we start checking **Step 5** as an example:

*Solution:*
*(Step 0) Let $z = x + yi$ be a complex number, where $x$ and $y$ are real numbers.*
*(Step 1) We are given that the real part of $\frac{1}{z}$ is equal to $\frac{1}{6}$.*
*(Step 2) Using the fact that the real part of a complex number $a + bi$ is $a$, we have*
$\operatorname{Re}\left(\frac{1}{z}\right) = \operatorname{Re}\left(\frac{1}{x+yi}\right) = \frac{x}{x^2+y^2}$.
*(Step 3) Setting $\operatorname{Re}\left(\frac{1}{z}\right) = \frac{1}{6}$, we get $\frac{x}{x^2+y^2} = \frac{1}{6}$.*
*(Step 4) Cross-multiplying and rearranging, we get the equation $6x = x^2 + y^2$.*
*(Step 5) Completing the square, we obtain $\left(x - \frac{3}{2}\right)^2 + y^2 = \frac{9}{4}$.*
*(Step 6) This is the equation of a circle with center $\left(\frac{3}{2}, 0\right)$ and radius $\frac{3}{2}$.*
*(Step 7) The area of the region inside the circle is $\frac{9\pi}{4}$. So the answer is $\frac{9\pi}{4}$.*

**Variable definition extraction**   This optional stage extracts the definitions of variables that appear in the reasoning chain. SelfCheck feeds the definitions to the regeneration stage to avoid LLM not understanding the meaning of a variable. The input to the LLM is:

*The following is the solution to a math question: Question: Let $S$ be the set of complex numbers $z$ such that the real part of $\frac{1}{z}$ is equal to $\frac{1}{6}$. This set forms a curve. Find the area of the region inside the curve.*

*Solution:*
*(Step 0) Let $z = x + yi$ be a complex number, where $x$ and $y$ are real numbers.*
*...*
*(Step 7) The area of the region inside the circle is $\frac{9\pi}{4}$. So the answer is $\frac{9\pi}{4}$.*

*Are there variables in the solution? If so, please list the definition of variable in the form of "1. variable is defined as...".*

The underlined parts are the type of question, the question itself and the steps in its solution, respectively.

The output from the LLM is:

*Yes. There are variables in the solution.*
*1. **$z$ is defined as a complex number of the form $x + yi$, where $x$ and $y$ are real numbers.**​*

The bold part is then saved to form a part of the input in the regeneration stage.

**Target extraction**   To get a brief and clear target of the current step, the input to the LLM is:

*The following is a part of the solution to the problem: Let $S$ be the set of complex numbers $z$ such that the real part of $\frac{1}{z}$ is equal to $\frac{1}{6}$. This set forms a curve. Find the area of the region inside the curve.*

*(Step 0) Let $z = x + yi$ be a complex number, where $x$ and $y$ are real numbers.*
*...*
*(Step 5) Completing the square, we obtain $\left(x - \frac{3}{2}\right)^2 + y^2 = \frac{9}{4}$.*

*What specific action does the step "Completing the square, we obtain $\left(x - \frac{3}{2}\right)^2 + y^2 = \frac{9}{4}$." take? Please give a brief answer using a single sentence and do not copy the steps.*

The underlined parts are the question and reasoning steps before the current one, including the current one.

The output of the LLM is:

*The step completes the square to rewrite the equation in standard form of a circle.*

The whole sentence is saved and forms the most important part of the input in the regeneration stage.

**Information Collection**    To get sentences in the question and previous steps in the solution that are directly related to the current step, the input to the LLM is:

*This is a math question: Question: Let S be the set of complex numbers z such that the real part of $\frac{1}{z}$ is equal to $\frac{1}{6}$. This set forms a curve. Find the area of the region inside the curve.*

*The following is information extracted from the question:*
*Information 0: Let S be the set of complex numbers z such that the real part of $\frac{1}{z}$ is equal to $\frac{1}{6}$.*
*Information 1: This set forms a curve.*
*Information 2: Find the area of the region inside the curve.*

*The following is the first a few steps in a solution to the problem:*
*Step 0: Let $z = x + yi$ be a complex number, where x and y are real numbers.*
*...*
*Step 4: Cross-multiplying and rearranging, we get the equation $6x = x^2 + y^2$.*

*Which previous steps or information does the next step "Completing the square, we obtain $\left(x - \frac{3}{2}\right)^2 + y^2 = \frac{9}{4}$." directly follow from?*

The underlined information (contents after Information X:) is simply a list of sentences in the question. The underlined steps are reasoning steps before (not including) the current one in the solution.

The output from the LLM is:

*The next step "Completing the square, we obtain $\left(x - \frac{3}{2}\right)^2 + y^2 = \frac{9}{4}$." directly follows from **Step 4**.*

Then we use regular expression to extract the information and step id from LLM's output. Some steps may only rely on information from the question while others may only use results from previous steps.

**Step regeneration**    In this stage, we instruct the LLM to achieve the extracted target with only collected information with the following prompt:

*We are in a process of solving a math problem.*

*Variables are defined as: 1. z is defined as a complex number of the form $x + yi$, where x and y are real numbers.*

*The following are some previous steps: Step 0: Cross-multiplying and rearranging, we get the equation $6x = x^2 + y^2$.*

*The target for next step is: The step completes the square to rewrite the equation in standard form of a circle.*

*Please try to achieve the target with the information from previous steps.*

The underlined parts are the type of question, variable definition (if any), collected information from the question (if any), collected previous steps (if any) and the extracted target. Please notice that the current step does not directly rely on any information from the question, we removed the corresponding block from the input.

The output from the LLM is:

*To rewrite the equation in standard form of a circle, we need to complete the square for both x and y.*
*Starting from the given equation: $6x = x^2 + y^2$.*
*We add and subtract $(6/2)^2 = 9$ on the right-hand side:*
*$6x + 9 - 9 = x^2 + y^2 + 9 - 9$.*
*Simplifying: $(x - 3)^2 + y^2 = 9$. This is now in the standard form of a circle with center $(3, 0)$ and radius 3.*

Then we can directly compare the regenerated step with the original one in the next stage.

**Result comparison**    The input to the LLM is:

*The following are 2 solutions to a math problem:*

*Solution 1: To rewrite the equation in standard form of a circle, we need to complete the square for both $x$ and $y$.*
*Starting from the given equation: $6x = x^2 + y^2$.*
*We add and subtract $(6/2)^2 = 9$ on the right-hand side: $6x + 9 - 9 = x^2 + y^2 + 9 - 9$.*
*Simplifying: $(x-3)^2 + y^2 = 9$. This is now in the standard form of a circle with center $(3, 0)$ and radius $3$.*

*Solution 2: Completing the square, we obtain $\left(x - \frac{3}{2}\right)^2 + y^2 = \frac{9}{4}$.*

*Compare the key points from both solutions step by step and then check whether Solution 1 "supports", "contradicts" or "is not directly related to" the conclusion in Solution 2. Pay special attention to difference in numbers.*

The underlined parts are simply the problem type, the regenerated step and the original step.

The output is:

*In both solutions, the goal is to rewrite the given equation in standard form of a circle. However, in Solution 1, the completion of the square results in the center of the circle being $(3, 0)$ and a radius of $3$. In Solution 2, the completion of the square results in the center of the circle being $(3/2, 0)$ and a radius of $3/2$.*

*Therefore, Solution 1 **contradicts** the conclusion in Solution 2 as the centers and radii of the circles are different.*

If the word 'support' or 'contradict' appears in the last line of the output, SelfCheck outputs a step score 1 or -1. Otherwise, SelfCheck outputs 0 meaning that two steps are not related.

## APPENDIX B    EXAMPLAR FOR ONE-SHOT ERROR CHECKING

The following example from Ling et al. (2023) is used to perform one-shot error checking.

*Here is some information:*
*"1. Alice gets 2 apples.*
*2. Alice gets twice as many oranges as bananas.*
*3. Original price per apple: $2*
*Discount: 50%*
*Price per apple after discount: $2 * (1 - 0.50) = $2 * 0.50 = $1"*

*Based on the given information, here is a reasoning process:*
*"Calculate Alice's cost of the 5 apples after a 50% discount.*
*Price per apple after discount: $1*
*Apples be eaten: 3*
*Oranges be eaten: 6*
*Total apples: 5*
*Total oranges: x*
*Total bananas: 2x*
*Alice's total number of fruits: 5 + x + 2x = 5 + 4x*
*Alice's total cost of 5 apples: $1 * (5 - 3) = $2"*

*Double-check the reasoning process, let's analyze its correctness, and end with "yes" or "no".*

*Answer:*
*Let's think step by step.*

*Grounding check*
*Purpose: Find references for numbers in the reasoning process sequentially*
*Rule: Any ungrounded number makes the check fails*

*"x" is a variable, we do not care about variables.*
*"y" is a variable, we do not care about variables.*

*"5" is a quantity from the given information that represents the number of apples.*
*"50%" is a quantity from the given information that represents the discount.*
*"$1" is a quantity from the given information that represents the price per apple after the discount.*
*"3" is a quantity not from the given information, it is ungrounded.*
*"6" is a quantity not from the given information, it is ungrounded.*
*"$2" is a quantity from the given information that represents the original price per apple.*
*Result: Based on the check, there are ungrounded numbers, so the grounding check fails.*

*Reasoning check:*
*Purpose: Check whether the reasoning logic correctly answers the question*
*Rule: Any logic error makes the check fails*

*To calculate the total cost of apples after a 50% discount, we should multiply the number of apples by the discounted price. But to answer the question, we do not need to calculate the total number of fruit Alice gets.*
*Result: Based on the check, there are some logic mistakes, so the reasoning check fails.*

*Calculation check:*
*Purpose: Check whether the calculation process is consistent*
*Rule: Any inconsistent calculation makes the check fails*
*calculation1:*

*equation: $1 * (5 - 3), answer: $2*
*(5 - 3) = 2*
*$1 * 2 = $2 is consistent with the answer, so the calculation is correct.*
*calculation2:*
*equation: 5 + x + 2x, answer: 5 + 4x*
*x + 2x = 3x*
*5 + 3x is inconsistent with the answer, so the calculation is incorrect.*
*Result: Based on the check, the calculation process is inconsistent, so the calculation check fails.*

*Check results: Ground check fails, Reasoning check fails, Calculation check fails.*
*Rule: Any failed check makes the reasoning incorrect.*
*So the answer is "no".*

## APPENDIX C   MORE EXAMPLES ON TARGET EXTRACTION

The following are some real examples of extracted targets on MathQA with GPT-3.5.

*Step: According to the problem, the profit was divided in the ratio of 2:3, which means that x received 2/5 of the profit and y received 3/5 of the profit.*
*Extracted target: It calculates the ratio of the profit earned between x and y based on the information given in the problem.*

*Step: Since the interest has increased by 4%, the new rate of interest (r2) is 16.67% + 4% = 20.67%.*
*Extracted target: The step calculates the new rate of interest after increasing it by 4%.*

*Step: The sum of the ages of the first 14 students is 56 + 160 = 216 years.*
*Extracted target: The step calculates the sum of the ages of the first 14 students of the class.*

*Step: We can rearrange 8a = 9b to get a/b = 9/8.*
*Extracted target: The step rearranges the given equation to express a in terms of b and then obtains the ratio of a to b.*

*Step: So, the equation becomes: 33 + (1+y/100)\*50 = 83.*
*Extracted target: The step sets up the equation to calculate the percentage increase in Shyam's weight.*

*Step: Find the ratio of investment of each person in the business*
*Investment of A : Investment of B : Investment of C*
*= 6500 : 1300 : 7800*
*= 5 : 1 : 12*
*Extracted target: The step finds the ratio of investment of each person in the business.*

## APPENDIX D   AN EXPERIMENT ON LOGIC REASONING TASK

In this section, we verify whether SelfCheck can generalize to the task of logic reasoning using the dataset of SNR.[2] Each test sample of SNR consists of a logic question, as well as its target answer. Different from other datasets shown in the paper, the target answers of SNR samples are natural-language sentences (such as "The hedgehog is cold and scary."), instead of multiple-choice options (for MathQA) or single math expressions (for GSM8K and MATH). Because of the variability of natural languages, exact matching among generated answers becomes infeasbile. Majority voting thus cannot be used for this problem, but we can still use SelfCheck to pick out the best solution from all candidates.

To test performance, we randomly select 100 test samples from the SNR test set and generate 2 answers for each of the questions. Because of the nature of natural language answers, it is not possible to perform automatic comparison between the generated answers and the target answers, so we invited volunteers to perform human evaluation. To better deal with logic problems, we replaced all 'math' with 'logic' in the prompts and deleted the instruction 'pay special attention to the difference in numbers' in the result comparison stage. The prompts used were otherwise identical to those outlined in Section 3.

From the results shown in Table D.1, we can see that SelfCheck increases the accuracy on SNR by 3.5%, which is a statistically significant improvement at 5% level.

Table D.1: SelfCheck increases predictive accuracy on SNR by picking out correct answers from two independent generations. Base Acc means single solution accuracy here, since majority voting is not feasible for SNR.

| Dataset | Generator | Checker | Base Acc (%) | SelfCheck Acc (%) | $\Delta$Acc (%) |
|---------|-----------|---------|--------------|-------------------|------------------|
| SNR | GPT-3.5 | GPT-3.5 | 44.5 | 48.0 | 3.5$\pm$2.0 |

## APPENDIX E   AN EXPERIMENT ON LLAMA2

To test SelfCheck's generalization to other LLMs, we also tested it with Llama2 (70B, 4-bit, Touvron et al. (2023)) on GSM8K (the generative accuracy of Llama2 on the other datasets was too low to perform any meaningful checking). Restricted by the speed of Llama2 (70B, 4-bit) on our server (which is only  20 tokens/s), we randomly select 500 test samples and generate 3 solutions for each question.

From Table E.1, we see a statistically significant improvement in accuracy compared with majority voting using this different LLM; the gains here are actually larger than those observed for GPT-3.5.

Table E.1: SelfCheck significantly increases predictive accuracy on GSM8K with Llama2.

| Dataset | Generator | Checker | MV Acc (%) | SelfCheck Acc (%) | $\Delta$Acc (%) |
|---------|-----------|---------|------------|-------------------|------------------|
| GSM8K | Llama2 | Llama2 | 40.3 | 43.2 | 2.9$\pm$0.3 |

---

[2] https://huggingface.co/datasets/lighteval/synthetic_reasoning_natural/blob/main/hard/test.jsonl

