# OpenReview forum: "SelfCheck: Using LLMs to Zero-Shot Check Their Own Step-by-Step Reasoning"
_ICLR.cc/2024/Conference — ICLR 2024 poster_

### Official Review · Reviewer_6G8r · 2023-10-31

**Soundness:** 3 good
**Presentation:** 4 excellent
**Contribution:** 3 good
**Rating:** 8
**Confidence:** 4

**Summary:**

This paper proposes a Self-Check, a method that allows LLMs to check their reasoning. The method involves four steps (target extraction, information collection, step regeneration, and result comparison.), each involving querying LLM for a different step. The basic idea involves probing the model to understand what a substep is doing (Steps 1 and 2), and then solving a substep again with this increased understanding (Step 3). If the generated answer does not match the conclusion of the subset, it indicates a contradiction in the output at the step. The contradictions are accumulated to generate a single score for the correctness of the solution. During inference, multiple solutions are generated, and SelfCheck scores are used to weigh and pick the best solution. This score is used to weigh solutions. The authors also analyze whether the SelfCorrect predictions correspond to the actual correctness and find that it is generally the case.

**Strengths:**

**A:** The method is straightforward yet proves to be effective. Although there are questions regarding its efficiency and effectiveness in comparison to majority voting (refer to weaknesses), the analysis confirms that SelfVerification is indeed taking place.

**B:** The paper demonstrates that LLMs can self-correct for reasoning problems in certain cases, providing a valuable answer to a significant question currently of interest to the community.

**Weaknesses:**

**A:** SelfCheck requires 3 LLM queries per substep. Consequently, the comparison between SelfCheck and SelfConsistency appears skewed, as SelfCheck inherently utilizes approximately 10x more samples (assuming an average of 3 substeps). Furthermore, the results with GPT-4 are based on a limited sample size, and given the analysis with Self-Consistency, the findings should be interpreted cautiously.


**B:** While the method is presented as 0-shot, it requires training samples (as seen in section 3.2) to learn hyperparameters, and thus, the criticism of approaches that require examples is unfair (We note though that these approaches are not directly comparable to SelfCheck in general, as they require additional exemplars which will often not be available in practice.). Additionally, the authors had to meticulously craft instructions for each step, which somewhat undermines their critique of other methods.

**Questions:**

Q1. If my observations in Weakness A are accurate, could Figure 5 simply result from comparing incompatible data points? Should we compare SelfCheck @ k with SelfConsistency @ m with m ~ 10 x k?

Q2: What are the false positive rates at various thresholds?

Q3: When devising the prompts, a small number of training samples from MathQA dataset were utilized. How were these examples used?


---

### Comments

- _Therefore, with SelfCheck, LLMs can effectively rectify their own biased beliefs by themselves._ While this insight is crucial, it might not be immediately apparent to a casual reader. In my opinion, this is an important takeaway from the paper. A more extensive analysis of this topic would be highly beneficial.

- Missing citation:

Jiang, Weisen, Han Shi, Longhui Yu, Zhengying Liu, Yu Zhang, Zhenguo Li, and James T. Kwok. "Backward reasoning in large language models for verification." arXiv preprint arXiv:2308.07758 (2023).

---

> ### Author Response · Authors · 2023-11-20
> **Response to Reviewer 6G8r**
>
> Thanks for your kind review and useful suggestions! Please let us know if the following answers your questions and addresses your concerns, we are very happy to provide additional information if needed.
>
> &nbsp;
>
>
> >* (Weakness A) SelfCheck requires 3 LLM queries per substep. Consequently, the comparison between SelfCheck and SelfConsistency appears skewed, as SelfCheck inherently utilizes approximately 10x more samples (assuming an average of 3 substeps). Furthermore, the results with GPT-4 are based on a limited sample size, and given the analysis with Self-Consistency, the findings should be interpreted cautiously.
>
> Thanks for the great question! We have a discussion on the cost of SelfCheck at the end of the general response. The cost of SelfCheck is at most 2x the cost generator on GPT-3.5 (as the cost is calculated based on tokens rather than api calls) and we think that the benefits of using SelfCheck will often justify its cost. For example, Figure 5 shows that majority voting can quickly saturate beyond a small number of solutions while SelfCheck does not, while the new Appendix D shows there are cases where majority voting cannot be used but SelfCheck can.
>
> Moreover, as discussed at the start of the general response, the utility of SelfCheck extends beyond just improving final predictive accuracy.  As you say, showing that LLMs can check their answers and rectify their own biases is of significant finding in itself, with various practical benefits of its own.
>
>
> &nbsp;
>
>
> >* (Weakness B) While the method is presented as 0-shot, it requires training samples (as seen in section 3.2) to learn hyperparameters, and thus, the criticism of approaches that require examples is unfair
>
> SelfCheck is a real zero-shot method, and the hyper-parameters $\lambda_{-1}$ and $\lambda_{0}$ are fixed rather than learned from training samples: they are constant across all experiments and were chosen based on a small amount of upfront testing, rather than full hyperparameter tuning.  In particular, this initial testing suggested any reasonable values for $lamda_{-1}$ that are significantly higher than $lamda_0$ worked, so we never attempted to learn or directly tune them in practice.
>
> &nbsp;
>
>
> >* (Weakness B) Additionally, the authors had to meticulously craft instructions for each step, which somewhat undermines their critique of other methods.
> >* (Q3) When devising the prompts, a small number of training samples from MathQA dataset were utilized. How were these examples used?
>
> Our instructions are very generic and kept the same across experiments, rather than being customized to different problems. We believe this significantly distinguishes it from other work, where the examples used are generally problem-specific.
>
> When designing these, we used only 5 examples from MathQA and simply checked whether GPT-3.5 followed the instructions in our prompts and slightly adjusted the prompts if not. We did not do any direct tuning of the instructions beyond this.
>
>
> &nbsp;
>
> >* (Q1) If my observations in Weakness A are accurate, could Figure 5 simply result from comparing incompatible data points? Should we compare SelfCheck @ k with SelfConsistency @ m with m ~ 10 x k?
>
> That’s a great question! We continued running the majority voting baseline on MathQA until the solution number reached 200. However, the observed saturation of majority voting continues, with accuracy continuing to vibrate around 0.78~0.79, which is the accuracy achieved by SelfCheck with 5 solutions and about 0.07 points lower than SelfCheck when the number of solutions is 30. As the cost of each SelfCheck verification step is about double that of a generation step here, it means that it is outperforming in a cost-normalized manner once it uses 5 or more solutions.
>
> We would also again like to reiterate here that majority voting cannot always be used (e.g. the new experiment in Appendix D) and that SelfCheck provides benefits beyond final predictive performance.
>
>
>
>
> &nbsp;
>
>
> >* (Q2) What are the false positive rates at various thresholds?
>
> The false positive rates at various thresholds are:
>
> |&nbsp; &nbsp;&nbsp; $t$|0.0|0.5|0.6|0.7|0.8|0.9|0.93|0.96|0.99|1.0|
> | :- | :-: |:-: |:-: |:-: |:-: |:-: |:-: |:-: |:-: |:-: |
> |GSM8K|1.0|0.98|0.94|0.87|0.71|0.49|0.31|0.31|0.31|0.0|
> |MathQA|1.0|0.99|0.96|0.89|0.70|0.38|0.20|0.08|0.08|0.0|
> |MATH|1.0|0.98|0.94|0.87|0.73|0.45|0.36|0.18|0.18|0.0|
>
>
> &nbsp;
>
> >* Missing citation: "Backward reasoning in large language models for verification."
>
> We have added this citation. Thanks!

---

> ### Comment · Reviewer_6G8r · 2023-11-21
> **Thanks for answering my questions**
>
> Thanks for taking the time to answer my questions.
>
>  > as the cost is calculated based on tokens rather than api calls
> > we think that the benefits of using SelfCheck will often justify its cost.
>
> - Makes sense. It will be good to add some more details on the cost in the next version.
>
>
> > That’s a great question! We continued running the majority voting baseline on MathQA until the solution number reached 200. However, the observed **saturation of majority voting continues**, with accuracy continuing to vibrate around 0.78~0.79
>
> Great, this is a very important result!
>
>
> > a small amount of upfront testing
>
> > Our instructions are very generic and kept the same across experiments, rather than being customized to different problems. We believe this significantly distinguishes it from other work, where the examples used are generally problem-specific.
>
> > When designing these, we used only 5 examples from MathQA and simply checked whether GPT-3.5 followed the instructions in our prompts and slightly adjusted the prompts if not. We did not do any direct tuning of the instructions beyond this.
>
>  These statements are still vague, and exactly how much work has to go into designing the prompts is unclear. A detailed discussion of exactly how much effort was spent on this will strengthen the paper.
>
> Since most of my concerns are addressed, I am raising my score to 8.

---

> > ### Author Response · Authors · 2023-11-22
> > **Thanks!**
> >
> > Thanks a lot for your quick reply! We are glad that you find most of your concerns addressed. We have added a discussion on the cost of SelfCheck in the updated submission and will make the details of the prompt designing process clearer in the next version.

---

### Official Review · Reviewer_NsFJ · 2023-10-31

**Soundness:** 2 fair
**Presentation:** 3 good
**Contribution:** 2 fair
**Rating:** 5
**Confidence:** 4

**Summary:**

The paper focused on the task of refining CoTs when applying LLMs to reasoning. The main contribution is to propose a self-checking solution on each step in the reasoning chain of CoTs one by one. The proposed operations include target extraction, information collection, step regeneration, and results comparison. The main tech employs some specifically designed prompts for powerful LLMs, such as GPT 3.5 and GPT-4. The experimental comparisons on three datasets (GSM8K, MathQA and MATH) show that the proposed method could outperform existing majority voting methods, self-verification, and deductive verification.

**Strengths:**

1) Correcting each reasoning step through self-checking is interesting.
2) Employing LLMs to perform self-checking based on designed prompts may be useful for some applications.
3) The experimental results on three open-accessed datasets show the effectiveness of the proposed solutions.

**Weaknesses:**

1) I think discussing how to prompt some "incantations" to the LLMs (especially for such black-box LLMs) has no insight into this community. The proposed prompts for each component in the self-checking process are well-designed, especially for GPT-serial LLMs. In my opinion, they could not be well extended to other LLMs, such as LLama or Claude.

2) Step-by-step checking may have the problem of error propagation.

**Questions:**

1) Why the authors do claim the solution is self-checking? In Table 1, the authors also employ different LLMs on the Generator and Checker.

2) Are the proposed checking components and prompts in each step useful for other LLMs?

3) In equation (1), why lamda_-1 is setting to 1 and lamda_0 to 0.3? How to determine such parameters？

4) Why there is no dot for DV and SV in Figure 2 (b)

5) What is the meaning of the yellow curve in Figure 3?

6) I think that the discussion in subsection 5.2 is of no use. The performance heavily relies on the prompts. So I think if we adopt an optimal prompt in the global checking setting, better results may be obtained.

7) Why were the experiments conducted on the math-related tasks? I wonder where the performance goes when other types of reasoning are explored, such as logical reasoning, deductive reasoning, and temporal reasoning.

---

> ### Author Response · Authors · 2023-11-20
> **Response to Reviewer NsFJ (1/2)**
>
> Thanks for reviewing our paper! As explained in the general response, we have added 2 experiments as per your suggestions. Please let us know if the following answers your questions and addresses your concerns, we are very happy to provide additional information if needed.
>
> &nbsp;
>
>
> >* (Weakness 1) I think discussing how to prompt some "incantations" to the LLMs (especially for such black-box LLMs) has no insight into this community. The proposed prompts for each component in the self-checking process are well-designed, especially for GPT-serial LLMs. In my opinion, they could not be well extended to other LLMs, such as LLama or Claude.
>
> We have added an extra experiment using  Llama2 (70b) on GSM8K, which shows that SelfCheck generalizes well: we actually see a bigger gain from using SelfCheck than for GPT-3.5.
> Please see Q2 in the general response and the new Appendix E for more details. We unfortunately do not have access to Claude at the moment so cannot test it.  We find that the accuracy of the generator when using Llama2 is too poor to provide any meaningful results on MathQA and MATH (as almost all generations are incorrect).
>
> More generally, we strongly disagree that the contribution of our work is in prompting “incantations”.  The key aspect of SelfCheck is in its novel decomposition of the step verification problem, with the precise prompts used very generic and not finely tuned to the particular setup (e.g. the prompts are held fixed across datasets).  Showing that LLMs are capable of checking their responses is also of significant scientific interest in its own right.
>
>
> Finally, the vast majority of  LLM works involve prompt designing, including the very famous Chain-of-Thought~(CoT) paper [1], which is purely a prompt designing work but has significantly influenced the community.  We thus do not feel this would be reasonable grounds for rejecting the paper, even if it were a fair characterization.
>
> [1] Chain-of-Thought Prompting Elicits Reasoning in Large Language Models. Wei et al.
>
>
> &nbsp;
>
>
> >* (Weakness 2) Step-by-step checking may have the problem of error propagation.
>
> We feel there may have been a significant misunderstanding here: though error propagation can indeed be an issue with CoT reasoning itself, SelfCheck is itself a mechanism to catch such errors and does not suffer from propagation of errors in its own verifications. Note here that SelfCheck verifies each reasoning step *independently*, conditional on information gathered from its context. If an error in our verification happens during verification of one step, it won’t affect other verification steps as the results of earlier verifications are not used in the verification process of later steps.

---

> > ### Comment · Reviewer_NsFJ · 2023-12-04
> >
> > Thanks for your replies. My replies and comments are listed below.
> >
> > For the weakness 1, I appreciate that the experiments on llama were added. However, my concern is that most current work on prompt engineering seems to work well on GPT series models. So, could it be beneficial to other kinds of foundation models? CoT is the first work on prompt designing. In this way, I think its contributions are well.
> >
> > For the weakness 2, If there is an error in the verification step, how we ensure such error cannot affect other steps?

---

> ### Author Response · Authors · 2023-11-20
> **Response to Reviewer NsFJ (2/2)**
>
> >* (Question 1) Why the authors do claim the solution is self-checking? In Table 1, the authors also employ different LLMs on the Generator and Checker.
>
> We focus on the key idea that an LLM can check its own reasoning because we believe that this is of scientific interest in itself, while most of our experiments are using the same LLMs as the generator and the checker. The cross-checking ability is an extension to this self-verification ability which might often be of additional practical use, but does not detract from our main contribution.
>
> &nbsp;
>
> >* (Question 2) Are the proposed checking components and prompts in each step useful for other LLMs?
>
> Yes. The prompts were designed with GPT-3.5 on 5 examples from MathQA, but were used in every experiment in the paper. Transfer has already been shown between GPT 3.5 and GPT 4, as well as different datasets. As previously mentioned, we have also done an extra experiment to verify the generalization ability of SelfCheck to Llama2 and found strong performance.
>
> &nbsp;
>
>
> >* (Question 3) In equation (1), why lamda_-1 is setting to 1 and lamda_0 to 0.3? How to determine such parameters？
>
> The hyperparameters $\lambda_{-1}$ and $\lambda_{0}$ are simply introduced to reflect the belief that 'contradict' steps affect the overall correctness of a reasoning chain more than 'is not related' steps.  Initial experiments showed that any positive $\lambda_{0}$ and a significantly larger $\lambda_{-1}$ (for example, $\lambda_{-1}$=1.0, $\lambda_{0}$=0.3) worked similarly well.  We, therefore, fixed them to these values for all experiments rather than individually tuning them, highlighting the robustness and generalisation of our approach.
>
>
> &nbsp;
>
>
> >* (Question 4) Why there is no dot for DV and SV in Figure 2 (b)
>
> The plotted values are taken directly from the reported results in the respective DV and SV papers, neither of which provided results for MathQA.  Unfortunately, due to budget restrictions, we were unable to rerun their approaches to provide our own values (and thus provide results for them on MathQA).
>
>
> &nbsp;
>
> >* (Question 5) What is the meaning of the yellow curve in Figure 3?
>
> It's the proportion of real wrong solutions in solutions that are predicted to be wrong. We have made it clearer in the paper.
>
> &nbsp;
>
> >* (Question 6) I think that the discussion in subsection 5.2 is of no use. The performance heavily relies on the prompts. So I think if we adopt an optimal prompt in the global checking setting, better results may be obtained.
>
> We believe that our results directly provide strong evidence against this assessment.  This section is an ablation to try and establish if a global checker can achieve similar performance with the right prompt, and we emphatically find that this is not the case. Note that we carefully tried to tune this global prompt to try and make it effective (this is actually how we started the project and the idea of the SelfCheck approach came out of the fact that we could not make it work). Others have also similarly presented results suggesting this does not work, see e.g. Table 2 in [2].
>
>
> [2] Deductive verification of chain-of-thought reasoning. Ling et al.
>
> &nbsp;
>
> >* (Question 7) Why were the experiments conducted on the math-related tasks? I wonder where the performance goes when other types of reasoning are explored, such as logical reasoning, deductive reasoning, and temporal reasoning.
>
> This is a fair question and we agree that experiments on other tasks will improve the paper.  To this end, we have performed an extra experiment on logic reasoning using the SNR dataset. Please find the results in the general response, Q1, and the new Appendix D in the updated paper. We find that SelfCheck again provides benefits in this setting.
>
> We note here that the original focus on math-based datasets is in line with most other works in the literature, and was primarily based on dataset availability and ease of performance evaluation.  For example, our new experiment required volunteers to manually evaluate the answers.

---

> > ### Comment · Reviewer_NsFJ · 2023-12-04
> >
> > I have read your replies. Thanks for your clarification.

---

### Official Review · Reviewer_PScZ · 2023-11-01

**Soundness:** 3 good
**Presentation:** 3 good
**Contribution:** 3 good
**Rating:** 6
**Confidence:** 4

**Summary:**

This paper proposes a zero-shot verification schema named SelfCheck to recognize errors in the step-by-step reasoning of large language models (LLMs) in answering complex questions. The proposed SelfCheck implements the self-checking by following a regenerate-and-compare procedure and then conducts weighted voting on multiple solutions, thereby improving the performance. Experimental results on three maths datasets including GSM8K, MathQA and MATH demonstrate the effectiveness of SelfCheck, which alleviates the need for additional data or external resources.

**Strengths:**

1. This paper explores the self-checking capabilities of LLMs by elaborating a regenerate-and-compare schema to assess their reasoning and recognize errors, which potentially has broader insights beyond a wider range of tasks.

2. The authors introduce an intriguing and reasonable ensemble method for multiple solutions, which employs an integration function and calculates confidence scores in the final voting stage.

3. The authors provide a comprehensive and detailed analysis to validate the effectiveness of each stage of the proposed SelfCheck.

**Weaknesses:**

1. The authors claim that they provide a general-purpose, zero-shot approach to checking. However, the proposed schema is more tailored to maths problems to some extent as answering mathematical questions requires explicit computational steps and numerical formulas, which can be easily checked by LLMs. Hence, it is a little confusing to state the proposed SelfCheck is general-purpose. I wonder if the authors could conduct experiments on other types of reasoning tasks such as logical reasoning or commonsense reasoning.

2. Confusion about the methodology design.

- The authors check each step by evaluating the conditional correctness of each step based on the provided question and previous steps in the solution. I doubt whether there could be a phenomenon where step i is wrong, but the following steps are perfectly correct, i.e. the model is anot affected by the step i error. But if the authors evaluate using conditional errors, it’s quite possible that the model would determine all subsequent steps (after step i) are wrong. Could the authors provide more analysis or explanations?

- The design of the integration function seems helpful as shown in the experimental results. Could the authors explain the insights of this design and the choices of hyperparameters?

3. A lack of clarification regarding the experimental results.

- The authors claim that SelfCheck significantly outperforms majority voting with both GPT-3.5 and GPT-4. However, results in Table 1 (Row 2) show that SelfCheck exhibits inferior performance in GSM8K with GPT-4 as the generator and checker.

- Results in Figure 2 demonstrate that the SelfCheck did not exceed majority voting by much, but the inference cost of SelfCheck should be larger in comparison. Could the authors illustrate in detail the superiority of SelfCheck over majority voting?

4. The writing can be improved. There are some typos and unclear descriptions. Please refer to the questions below for details.

**Questions:**

1. In Paragraph 2, Section 3.1, the authors state that ‘checking’ is a less common task in the training corpus of most LLMs. Can the authors provide some evidence of citations of this statement?

2. In Paragraph 5, Section 3.1, the authors claim that LLM is usually able to perform the tasks of figuring out the target of the current step. Missing evidence or supplementary experiments should be added.

3. In Section 3.1, the authors mention Information in both stages of Information collection and Step regeneration but the Information should refer to different contents according to the authors’ descriptions. A minor suggestion would be to modify the terms in the prompts in order to better distinguish them.

4. The comparison between the proposed SelfCheck and DV/SV in Figure 2 is quite confusing. Could the authors give relevant explanations on how they compare these methods with varying scores and different generator models?

5. Why did not the authors conduct experiments with GPT-3.5 as the Generator and GPT-4 as the Checker (in Table 1)? It would be helpful to provide appropriate descriptions.

6. Minor comments on writing:

(1)	Paragraph 2 in Section 1: ..., which is far below human level -> , which  is far below the human level

(2)	Paragraph 5 in Section 1 & Paragraph 1 in Section 3: to check their working -> check their work

(3)	Paragraph 1 in Section 3.2: , and thus final solution -> , and thus the final solution

[1] Jie Huang, Xinyun Chen, Swaroop Mishra, Huaixiu Steven Zheng, Adams Wei Yu, Xinying Song, Denny Zhou. LARGE LANGUAGE MODELS CANNOT SELF-CORRECT REASONING YET

---

> ### Author Response · Authors · 2023-11-20
> **Reply to Reviewer PScZ (1/3)**
>
> Thanks a lot for your detailed and helpful review! We have added an experiment and updated the paper as per your suggestions. Please let us know if the following answers your questions and addresses your concerns, we are very happy to provide additional information if needed.
>
> &nbsp;
>
>
> > (Weakness 1) I wonder if the authors could conduct experiments on other types of reasoning tasks such as logical reasoning or commonsense reasoning
>
> This is a great suggestion that we think will improve the paper. We’ve added an experiment on logic reasoning with the dataset of SNR, which shows the generalization ability of SelfCheck to non-math reasoning problems. We find that SelfCheck is effective in this setting as well. Please see Q1 in the general response and the new Appendix D for details.
>
> &nbsp;
>
> > (Weakness 2) I doubt whether there could be a phenomenon where step i is wrong, but the following steps are perfectly correct, i.e. the model is not affected by the step i error. But if the authors evaluate using conditional errors, it’s quite possible that the model would determine all subsequent steps (after step i) are wrong. Could the authors provide more analysis or explanations?
>
> Thank you for raising this, we may have not made it sufficiently clear what we mean by the 'conditional correctness' of a step previously. The evaluation of each step is evaluated independently and does not depend on whether previous errors were detected: we assess correctness conditioned on taking the information gathered from context to be true.  We find many cases where subsequent steps are conditionally correct extensions of previous erroneous steps. They do not have to reach a correct final conclusion for conditional correctness to hold; indeed miraculous corrections back to the true correct answer are much rarer.
>
> To give a concrete example, if we had the question “What is 4x(3+2)?” and Step 1 concluded that 3+2=6, then it would be conditionally correct for Step 2 to conclude that 4x(3+2)=24, conditional on the information in context that 3+2=6.
>
> We have clarified this in the updated submission.
>
>
> &nbsp;
>
> > (Weakness 2) The design of the integration function seems helpful as shown in the experimental results. Could the authors explain the insights of this design and the choices of hyperparameters?
>
> In order not to introduce extra complexity, the integration function is simply a linear model.
> The hyper-parameters $\lambda_{-1}$ and $\lambda_{0}$ are fixed across all experiments and were chosen based on a small amount of upfront testing, rather than full hyperparameter tuning. The introduction of the hyper-parameters $\lambda_{-1}$ and $\lambda_{0}$ is to instill the inductive bias that 'contradict' steps have more effect on the overall correctness than 'not related' steps into SelfCheck. In particular, this initial testing suggested any reasonable values for $lamda_{-1}$ that are significantly higher than $lamda_{0}$ worked, so we never attempted to learn or directly tune them in practice.
>
> &nbsp;
>
>
> > (Weakness 3) Results in Table 1 (Row 2) show that SelfCheck exhibits inferior performance in GSM8K with GPT-4 as the generator and checker.
>
> The results are very close and not statistically significant in this case. However, it is not immediately clear why SelfCheck is not helpful in this single configuration. We guess it might be because GPT-4 is already strong enough to solve the simple problems in GSM8K without being affected by its biases. We note that gains are significant in almost every other setting.

---

> ### Author Response · Authors · 2023-11-20
> **Reply to Reviewer PScZ (2/3)**
>
> > (Weakness 3) Results in Figure 2 demonstrate that the SelfCheck did not exceed majority voting by much, but the inference cost of SelfCheck should be larger in comparison. Could the authors illustrate in detail the superiority of SelfCheck over majority voting?
>
> We would first like to highlight here, as discussed at the start of the general response, that SelfCheck is more than a method for improving predictive accuracy. It is thus immediately beneficial over majority voting in showing the crucial insight that LLMs are capable of checking their own answers in a meaningful way and allowing tasks such as verification, confidence scoring, and filtering of individual answers.
>
> Second, we believe that there are important ways that SelfCheck can be better than majority voting in a larger ensemble from a purely predictive perspective:
> - As shown in Figure 5, the performance of majority voting saturates with the number of solutions.  It never does better than SelfCheck with 5 solutions, no matter how many solutions are used, with SelfCheck giving around a 7\% gain in performance for large ensembles.
> - There are many cases in which majority voting cannot be used because answers are in free text form rather than being numeric, see the new experiment in Appendix D.
>
> Finally, we would also like to point out that the performance gap between SelfCheck and majority voting is significant. Selfcheck increases accuracy by ~1/% to ~4\% compared with majority voting for all settings. By comparison, previous verification-based methods such as Deductive Verification and Self-Verification only provide an increase of at most 1\%.
>
>
> &nbsp;
>
>
> > (Weakness 4) There are some typos and unclear descriptions. Please refer to the questions below for details.
>
> Thank you for pointing them out; we have corrected them.
>
> &nbsp;
>
>
> > (Question 1) In Paragraph 2, Section 3.1, the authors state that ‘checking’ is a less common task in the training corpus of most LLMs. Can the authors provide some evidence of citations of this statement?
>
> Sorry this wasn't so clear in the paper. We are simply alluding to the fact that they are trained as generative models, instead of directly trained as methods for checking. The comparison in Section 5.2 shows the superiority of regeneration & comparison compared with direct error checking, which supports the argument that they are more suited to this generation task. We have made edits to make this clear.
>
> &nbsp;
>
>
> > (Question 2) In Paragraph 5, Section 3.1, the authors claim that LLM is usually able to perform the tasks of figuring out the target of the current step. Missing evidence or supplementary experiments should be added.
>
> We find that SelfCheck can extract step targets very accurately in practice relative to the regularity of errors in the checking itself. It’s difficult to directly evaluate this quantitatively, as there is no absolute measure of its correctness for the target extraction and even trying to do our own manual labelling becomes quite subjective.  However, we have added some randomly selected examples in Appendix C to show its typical behaviour.
>
> &nbsp;
>
>
> > (Question 3) In Section 3.1, the authors mention Information in both stages of Information collection and Step regeneration but the Information should refer to different contents according to the authors’ descriptions. A minor suggestion would be to modify the terms in the prompts in order to better distinguish them.
>
> We use the same terms as the “Informations” in step regeneration are always a renumbered subset of those in information collection.  Note here $I_0$, $I_1$, ... are being directly used to index from the original set in the information Collection stage. For example, if $I_0=2$ in step regeneration, Information 2 from the question is used, and renumbered as the 0-th information to be fed into the regeneration stage.
>
> &nbsp;
>
>
> > (Question 4) The comparison between the proposed SelfCheck and DV/SV in Figure 2 is quite confusing. Could the authors give relevant explanations on how they compare these methods with varying scores and different generator models?
>
> Because of budget constraints, it is not possible for us to repeat DV and SV by ourselves. Instead, we compare gains in accuracy over majority voting based on their reported results (noting that differences in generators mean that absolute accuracies are not comparable), matching the number of solutions used by each paper (10 for DV and 5 for SV). In both cases that we also test DV is actually worse than performing no checking, while SV has a gain of less than 0.5\%.  More generally, both approaches only very rarely provided gains of more than 1\% over majority voting across their experiments. Thus the gains we have demonstrated for SelfCheck are noticeably bigger than those reported in any setting by those papers.

---

> ### Author Response · Authors · 2023-11-20
> **Reply to Reviewer PScZ (3/3)**
>
> > (Question 5) Why did not the authors conduct experiments with GPT-3.5 as the Generator and GPT-4 as the Checker (in Table 1)? It would be helpful to provide appropriate descriptions.
>
> The reason is that using GPT-4 to check GPT3.5 is not a reasonable setting in practice and our limited budget meant we needed to be selective on what was run. The default setting of SelfCheck is to use the same kind of LLMs to check themselves. We further added the potentially useful case of using a cheaper model (GPT-3.5) to check more powerful (and expensive) ones (GPT-4). Using a more powerful model to check a cheaper one doesn't make much sense, because directly using the more powerful model to generate solutions would usually result in better performance.

---

### Official Review · Reviewer_wFF7 · 2023-11-03

**Soundness:** 2 fair
**Presentation:** 2 fair
**Contribution:** 2 fair
**Rating:** 5
**Confidence:** 4

**Summary:**

The paper introduces SelfCheck, a zero-shot verification framework designed to identify errors in Large Language Models (LLMs) when applied to mathematical reasoning tasks. Instead of the simplistic approach of self-querying LLMs, the paper presents a multi-stage strategy that deconstructs the problem into a sequence of simpler tasks. This method capitalizes on the LLM's generative capabilities and reduces the correlation between errors in the original generation and the subsequent verification process.

The approach involves separate interactions with the LLM, where it extracts the target and relevant context for each step, then generates an independent alternative step based on this information. Subsequently, it compares the original step with the regenerated one. If they match, the original step passes the verification check. The approach also employs confidence scores as weights to evaluate and prioritize more accurate solutions, offering a flexible alternative to traditional majority voting methods.

Notably, this approach operates in a zero-shot manner, necessitating neither illustrations nor training.

**Strengths:**

Using the LLM to identify errors in its own step-by-step reasoning, analogously to how a human might go back to check their working is a very interesting research problem. Self checking and further refinement of its own answers is a challenging task for LLMs when reasoning is involved. The paper proposes a general-purpose, zero-shot, checking schema for self-identifying errors in LLM CoT reasoning. Some past methods have dealt with this problem but almost all of them use few shot setting. It is interesting to see the method work in zero shot setup.

The paper introduces an innovative step-checking method that leverages the generative modeling capabilities of modern LLMs to determine the correctness of step-by-step reasoning. To achieve this, the paper divides the checking process into four key components: target extraction, information gathering, step regeneration, and result comparison. Each step is carried out with carefully designed prompts, and the collective outcomes of these steps determine the overall accuracy of the answer.

**Weaknesses:**

- The paper presents a general purpose validation framework designed to assess the accuracy of multi-step mathematical reasoning tasks. Although this approach performs effectively well within a zero-shot prompting setup, it essentially relies on well-crafted prompts. It remains uncertain whether these four steps constitute the exclusive and comprehensive way of rigorously assessing the accuracy and the reasoning process in such tasks. There might be more ways to asses the quality of the reasoning chain and there is no gurantee that this is the only way or the best way. These four steps are not acquired through learning, generated by a model, nor rooted in learning science or based on any other theory. So it is very hard to judge it. It is plausible that their effectiveness is coincidental in the context of mathematical reasoning tasks and may not readily apply to other types of reasoning. Consequently, for broader applicability, the prompts would need to be rethought and rephrased, raising questions about the approach's generalizability.

- Also, I dont see the usefulness of zero shot setting as if this comprehensive prompt for verification can be written, then 1 or 2 examples can also be provided. Utilizing a few-shot configuration could have yielded improved outcomes and enhanced the model's ability to adhere to the prompt, potentially resulting in better overall performance. Moreover, in a few-shot setup, the approach could have been benchmarked against previous methods. If the intention behind the zero-shot setup was to demonstrate the broad applicability of the approach, it would have been much better to include datasets that covered a wider range of domains, rather than solely focusing on mathematical reasoning.

- The paper lacks a comparative analysis with other methods that have employed extensive prompts, such as PHP , faithful decomposition, verification based on subquestion decomposition, and similar approaches. So it is hard to judge if this prompting strategy is the best one for mathematical reasoning.

- No comparison has been drawn to the cost of the prompting method as for each sample, 4 extra verification steps are needed which makes 5 API calls (at least) per sample. Compared to one step verification tasks that take 2 API calls, is the cost vs accuracy tradeoff worth it?

- Table 1 is a bit confusing as 2 samples for majority voting does not make much sense. Should be at least 3 so that there are some agreements between the samples.

Papers:
[PHP]: https://arxiv.org/abs/2304.09797
[Faithful decomposition]: https://www-files.anthropic.com/production/files/question-decomposition-improves-the-faithfulness-of-model-generated-reasoning.pdf

**Questions:**

Questions are mentioned in the weakness section. Please refer to that.

---

> ### Author Response · Authors · 2023-11-20
> **Reply to Reviewer wFF7 (1/2)**
>
> Thank you for your helpful review! As detailed in the general response, we have added an experiment to show SelfCheck’s generalization ability to non-math reasoning problems. Please let us know if the following answers your questions and addresses your concerns, we are very happy to provide additional information if needed.
>
> &nbsp;
>
>
> > (This approach) essentially relies on well-crafted prompts… It is plausible that their effectiveness is coincidental in the context of mathematical reasoning tasks and may not readily apply to other types of reasoning.
>
> To address your concern, we have added an extra experiment on logic reasoning, which shows that SelfCheck generalizes well to non-mathematical reasoning tasks. Please see the response to Q1 in the general response and the new Appendix D in the updated paper.
>
> &nbsp;
>
> > It remains uncertain whether these four steps (in SelfCheck) constitute the exclusive and comprehensive way of rigorously assessing the accuracy and the reasoning process in such tasks. There might be more ways to asses the quality of the reasoning chain and there is no gurantee that this is the only way or the best way.
>
> SelfCheck shows that LLMs can effectively detect their own errors, which answers an important question in the LLM community and is an important contribution in its own right. We agree that the idea could potentially be adapted and built on further, but this does not mean it is not already useful. We actually view this as a strength of the work as we are opening up new avenues for research and feel it would be very counterproductive to reject the paper on this basis. Critically, we are not aware of any previous work that has achieved a comparable level of performance in checking LLM CoT reasoning (especially not in a zero-shot manner).
>
> &nbsp;
>
> > I dont see the usefulness of zero shot setting as if this comprehensive prompt for verification can be written, then 1 or 2 examples can also be provided.
>
> Perhaps the most important point to note here is that there are not currently any few-shot methods that provide the same level of performance as SelfCheck, see e.g. the performance of DV and SV in Figure 2.  The only methods that can do better rely on finetuning from large quantities of problem-specific examples.  Thus the utility of SelfCheck goes beyond its being zero-shot: it shows LLMs can meaningfully self-check without requiring access to large amounts of new data.
>
> Users of SelfCheck can certainly add few-shot examples to boost its performance on a specific task. However, we focus on the zero-shot setting because it provides significant practical advantages. Namely, being zero-shot makes SelfCheck generalisable and deployable, which can be done in a more automated back-end way. On the contrary, few-shot examples need to be designed separately for even slightly different datasets. For example, we found few-shot prompts that work well on GSM8K fail completely on MathQA and MATH in our preliminary experiments, but the same zero-shot instructions work for all three datasets.
>
>
>
> &nbsp;
>
> > The paper lacks a comparative analysis with other methods that have employed extensive prompts, such as PHP , faithful decomposition, verification based on subquestion decomposition, and similar approaches. So it is hard to judge if this prompting strategy is the best one for mathematical reasoning.
>
> Verification is itself an important question, so step-by-step verification of SelfCheck is useful in its own right. The aim of SelfCheck is checking and verification, which can help with downstream prediction, but this is not a paper rooted in simply trying to maximize final empirical predictive performance.
>
> We don’t compare to methods like PHP and faithful decomposition as they are aimed to directly increase predictive performance, but they are not capable of checking their own (and others) answers in a useful way. Moreover, they are not directly competing methods: SelfCheck can be used on top of them to check for their errors and further improve their performance.
>
> &nbsp;

---

> > ### Comment · Reviewer_wFF7 · 2023-12-01
> > **Response**
> >
> > I would thank the authors for the repsonse and all the experiments performed.
> >
> > 1. The experiment outside of the math domain is done on some synthetic dataset which is hard to compare against any method. I would have appreciated some popular dataset with a baseline to compare against.  Moreover, it is hard to know if the results are statistically significant.
> >
> > 2. I agree that finding the four steps is a great contribution but is there a way to know that these four steps are the most important ones? Is there a basis for that?
> >
> > 3. Again, my concern was not addressed here. I understand and completely agree that having zero shot capabilities is a good idea but showing 1 example as demonstration is not bad given that the prompts are designed carefully. So it would be a good idea to compare against that.
> >
> > 4. There are papers that cross exams each other answers like LM vs LM (https://arxiv.org/abs/2305.13281). Some baselines would be great to have.

---

> ### Author Response · Authors · 2023-11-20
> **Reply to Reviewer wFF7 (2/2)**
>
> > No comparison has been drawn to the cost of the prompting method as for each sample, 4 extra verification steps are needed which makes 5 API calls (at least) per sample. Compared to one step verification tasks that take 2 API calls, is the cost vs accuracy tradeoff worth it?
>
> Please see the response to Q3 in the general response. In short, we believe there are many settings where the benefits of using SelfCheck justify its cost. For example, Figure 5 shows that majority voting can quickly saturate beyond a small number of solutions while SelfCheck does not; the new Appendix D shows there are cases where majority voting cannot be used but SelfCheck can; SelfCheck can be used for verifying and filtering answers, rather than just to maximize predictive performance; and it can be used to identify the exact errors made in a CoT reasoning in an interpretable way.
>
> &nbsp;
>
> > Table 1 is a bit confusing as 2 samples for majority voting does not make much sense. Should be at least 3 so that there are some agreements between the samples.
>
> We agree that MV has a failure mechanism when only using 2 samples and so is essentially the same as the default prediction in this setting. We have made this clearer in the paper, highlighting that Figure 2 (and the new Tables D.1 and E.1) should be used as the main basis of comparison between SelfCheck and MV.
>
> What we want to show in Table 1 is that SelfCheck can pick out the right answer when different runs of the generator don’t agree with each other. For example, when the generator and checker are both GPT-3.5, SelfCheck has 12.2%, 20.7%, 9.1% higher probabilities to pick out the right answers for each dataset compared with random selection. This can also be useful in cases where the generator is no longer available, so we need to select the best solution from the solutions already generated.

---

### Author Response · Authors · 2023-11-20
**General Response**

We thank all reviewers for their time, insightful comments, and helpful feedback. We are glad to see that all reviewers agree that verification of reasoning is an important problem and that our method (SelfCheck) is generally regarded to be simple and effective. Below we clarify an important aspect of our contributions and answer some shared questions from the reviewers.  As part of the latter we provide results for two new experiments.

&nbsp;

> Utility of SelfCheck to literature


Reviewers generally focussed their assessments on the utility of SelfCheck around its ability to improve the final predictive performance of LLMs. We would like to highlight that we believe that its contributions go significantly beyond this. **SelfCheck demonstrates that LLMs can meaningfully detect errors in their own outputs.** We believe that this finding is of significant scientific merit in its own right, answering an important open question and providing key insight into their behavior; it is an important step towards building LLMs to generate reliable reasoning.

Improving predictive performance by weighted voting is only one of SelfCheck’s potential applications. Its success shows the effectiveness of SelfCheck in distinguishing correct reasoning chains from incorrect ones, but we believe this ability to self-check can also open up many avenues for future research, such as providing more accurate confidence estimates in answers and a mechanism to screen answers and detect hallucinations.


&nbsp;


> Q1. Does SelfCheck generalize to non-math reasoning?

This was an important question and we agree the paper will benefit from adding tests on it. To this end, we have added an experiment on the dataset of SNR [1] to evaluate SelfCheck’s performance on logic reasoning. The results below show that SelfCheck provides a statistically significant improvement over the base LLM accuracy (note that majority voting cannot be used for this problem, because answers are in natural language form so cannot be directly matched).  Please see Appendix D in the updated paper for full details.

| Dataset     | Generator | Checker | Base ACC(%)| SelfCheck ACC(%)|Difference (%)
| :- | :- | :- | :-: | :-: | :-: |
|SNR[1]|GPT-3.5|GPT-3.5|44.5|48.0|3.5$\pm$2.0|

[1] https://huggingface.co/datasets/lighteval/synthetic_reasoning_natural/blob/main/hard/test.jsonl

&nbsp;


> Q2. Does SelfCheck generalize to other LLMs?

This is another great question. To answer it, we added another experiment to evaluate SelfCheck’s performance on the open-source LLM Llama2 (70B, 4-bit) with the GSM8K dataset and 3 solutions per question.  As the table below shows, SelfCheck provides a significant improvement over majority voting in this setting; the gains are actually larger than those observed when using GPT-3.5.  Please see Appendix E in the updated submission for more details.


| Dataset     | Generator | Checker | Majority voting ACC(%)| SelfCheck ACC(%)|Difference (%)
| :- | :-  | :-  | :-:  | :-:  | :-:  |
|GSM8K|Llama2(70b)|Llama2(70b)|40.3|43.2|2.9$\pm$0.3|


&nbsp;


> Q3. Computation usage of SelfCheck compared with majority voting?

The cost of running the SelfCheck verifier is around twice that of the original generation (the exact cost can vary as it depends on the number of tokens, rather than the number of API calls);  we have added discussion on this to the paper.



However, as explained above, SelfCheck is not simply a method to replace ensembling, but a checking method that is useful in its own right. For example, SelfCheck provides a confidence score, which can help filter out erroneous answers and is important to high-stake decision-making. Also, SelfCheck points out the precise erroneous steps in a solution, which can be useful for interpreting and gathering insights into LLMs CoT reasoning.

Moreover, even if a user only cares about predictive accuracy, there are many cases where increasing the computation budget for majority voting does not work. In Figure 5, we see that the accuracy of majority voting saturates when the number of solutions per question exceeds 10. (We increased the solution number to 200 for majority voting, but the accuracy still doesn’t go up.) In comparison, SelfCheck continues to increase accuracy by correcting biases of the generator model.

Furthermore, there are many cases where majority voting cannot be used at all, such as our new logic reasoning experiments, because there is not a discrete set of possible answers. Through its confidence scores, SelfCheck provides a natural way to decide between different possible answers in such settings.

&nbsp;

Thanks again for all your hard work.  We hope our responses and update have addressed your key concerns, but please let us know if there are further changes you would like us to make.

---

### Meta-Review · Area_Chair_zmv6 · 2023-12-06

**Metareview:**

The paper introduces SelfCheck, a zero-shot verification framework aimed at detecting errors in Large Language Models (LLMs) when applied to mathematical reasoning tasks. Instead of relying on self-querying LLMs, it follows a regenerate-and-compare procedure and then conducts weighted voting on multiple solutions. Experimental results on three datasets (GSM8K, MathQA, and MATH) show that the proposed method outperforms existing methods, such as voting methods and self-verification. Additionally, the paper verifies that SelfCheck predictions generally align with actual correctness, further validating its utility. The paper has the following strengths:

1. Reviewers generally found the approach interesting and novel, and its step-by-step approach is analogous to how a human might go back to check their work.
2. The method is effective and relatively straightforward, making it applicable in a zero-shot setup.
3. The authors provide a comprehensive analysis to validate the effectiveness of each stage of the proposed SelfCheck.

Several of the reviewers' concerns were addressed during the discussion period (e.g., new experiments with new LLMs and another task), but a few weaknesses remain:

1. The paper lacks a comparison to related verification methods such as PHP and faithful decomposition. The authors responded that such a comparison would be difficult, but their justification doesn't seem very convincing, as other methods also use prompting, and it would be straightforward to replicate and compare against.
2. The experiments beyond the mathematics domain were conducted on a synthetic dataset. While this may be adequate to evaluate the methodology, it is less convincing in response to a request for experiments that demonstrate robustness on a wider range of *domains*.

Overall, the paper's approach is interesting and yields good results, but its empirical support is somewhat questionable due to the above two weaknesses. After discussion with the reviewers, the general agreement is that these two drawbacks are not fatal, and we have decided to accept the paper.

**Justification For Why Not Higher Score:**

Concerns about missing comparisons to methods such as PHP, and no evaluation across different domains.

**Justification For Why Not Lower Score:**

The verification approach of the paper is effective and interesting, while being zero-shot.

---

### Decision · Program_Chairs · 2024-01-16

Accept (poster)